# Clinical relevance for circulating cold-inducible RNA-binding protein (CIRP) in patients with adult-onset Still's disease

**Yuya Fujita**[1]*, **Toru Yago**[1], **Tomoyuki Asano**[1], **Haruki Matsumoto**[1], **Naoki Matsuoka**[1], **Jumpei Temmoku**[1], **Shuzo Sato**[1], **Makiko Yashiro-Furuya**[1], **Eiji Suzuki**[2], **Hiroshi Watanabe**[1], **Atsushi Kawakami**[3], **Kiyoshi Migita**[1]

**1** Department of Rheumatology, Fukushima Medical University School of Medicine, Fukushima, Japan, **2** Department of Rheumatology, Ohta-Nishinouchi Hospital, Koriyama, Fukushima, Japan, **3** Department of Immunology and Rheumatology, Division of Advanced Preventive Medical Sciences, Nagasaki University Graduate School of Biomedical Sciences, Nagasaki, Japan

* fujita31@fmu.ac.jp

## Abstract

**Data Availability Statement:** All relevant data are within the paper and its Supporting Information files.

### Background

Adult-onset Still's disease (AOSD) is a systemic autoinflammatory disease in which danger-associated molecular patterns (DAMPs)-mediated inflammasome activation seems to be involved in the disease pathogenesis. Cold-inducible RNA-binding protein (CIRP) belongs to a family of cold-shock proteins that respond to cellular stress and has been identified as a DAMP that triggers the inflammatory response. The aim of this study is to investigate the clinical significance of serum CIRP levels in AOSD.

### Methods

Serum samples were obtained from 44 patients with active AOSD or 50 patients with rheumatoid arthritis (RA), 20 patients with systemic lupus erythematosus (SLE), and 15 healthy control patients (HCs). Serum levels of CIRP and IL-18 were determined using enzyme-linked immunosorbent assay. Results were compared among AOSD patients, RA patients, SLE patients and HCs. Results were also analyzed according to the clinical features of AOSD.

### Results

Serum CIRP levels were significantly higher in AOSD patients compared with RA patients (median: 9.6 ng/mL, IQR [5.7–14.4] versus 3.2 ng/mL, IQR [1.9–3.8]; $p < 0.001$) and with HCs (2.8 ng/mL, [IQR; 1.4–4.9], $p < 0.001$). There was a significant positive correlation between serum CIRP levels and AOSD disease activity score (Pouchot's score $r = 0.45$, $p = 0.003$) as well as between AOSD-specific biomarkers ferritin and IL-18. However, there was no significant difference in the serum CIRP levels among AOSD patients with three different disease phenotypes.

**Funding:** The study was supported by grants from the Japan Grant-in-Aid for Scientific Research [20K08777](https://www.jsps.go.jp/english/e-grants/).

**Competing interests:** Kiyoshi Migita has received research grants from Chugai, Pfizer, and AbbVie outside the submitted work. Chugai, Pfizer, and AbbVie did not have any additional role in the study design, data collection and analysis, decision to publish, or preparation of the manuscript. Rest of the authors declares that they have no competing interests. This does not alter our adherence to PLOS ONE policies on sharing data and materials.

## Conclusions

These results suggest that CIRP may play a significant role in the pathophysiology of AOSD and could be a potential biomarker for monitoring the disease activity of AOSD.

## Introduction

Cold-inducible RNA-binding protein (CIRP) is a highly conserved 172-amino acid nuclear protein that belongs to the family of cold shock proteins [1]. CIRP functions as an RNA chaperon facilitating RNA translation and is ubiquitously expressed in various tissues [2]. CIRP is induced in response to stress, such as hypothermia, irradiation, and hypoxia, and is released into circulation resulting in cytokine induction by stimulating immune cells [3]. Whereas serum CIRP was undetectable or minimum in the HCs, its levels were elevated in the sera of patients with sepsis or organ-targeted ischemia [4]. In addition, increased tissue and serum levels of CIRP have been reported in several inflammatory diseases [5].

CIRP is expressed in various types of cells, including innate immune cells such as macrophages, neutrophils, epithelial, and endothelial cells [6]. CIRP is released as extracellular CIRP (eCIRP) from these cells by lysosomal exocytosis pathway and passively by cellular necrosis [7]. eCIRP is a proinflammatory molecule that triggers inflammatory responses and organ damage, suggesting a role for CIRP in immune responses and inflammatory pathways [8]. CIRP is important as a damage-associated molecular protein (DAMPs) and is implicated in inflammatory diseases [9]. For example, CIRP is involved in nuclear factor-κB (NF-κB) activation and the regulation of interleukin-1β expression [10].

Adult-onset Still's disease (AOSD) is a systemic inflammatory disorder of unknown etiology [11]. Although the pathogenesis of the disease is not fully clarified, inflammasome activation and subsequent IL-1β induction are involved in the disease pathogenesis in AOSD [12]. Inflammasomes are multimeric protein complexes that typically comprise a sensor for PAMPs and DAMPs [13]. Recent studies demonstrated that eCIRP is important as a DAMPs [9]. The involvement of CIRP in AOSD, however, is not well known. In this study, we hypothesized that the release of CIRP contributes to the inflammatory processes in AOSD through macrophage activation and subsequent cytokine production. Here we found that the serum levels of CIRP were elevated in patients with AOSD and correlated with disease activity.

## Materials and methods

### AOSD patients and controls subjects

A total of 44 untreated AOSD patients and 15 healthy controls (HCs) were included in the training and validation sets. All patients diagnosed with AOSD at the Department of Rheumatology, Fukushima Medical University Hospital from January 2005 to April 2020 were enrolled. All records were accessed from January 2020 to October 2020. Patients included had to be 17 years old or older to be diagnosed as AOSD according to the diagnostic criteria of Yamaguchi after exclusion of those with infectious, neoplastic, and autoimmune disorders [14]. We additionally collected serum samples from 8 treated AOSD patients at the time of remission in order to explore the longitudinal changes. As controls, 15 HCs (5 males, 10 females, median age 38 years, interquartile range [IQR]; 32–45 years) were included. HCs lacked chronic medical diseases or conditions and did not take prescription medications or over-the-counter medications within seven days. Additional independent sets consisting of 50

patients with rheumatoid arthritis (RA) and 20 patients with systemic lupus erythematosus (SLE) were used to determine the specificity of the values of CIRP in AOSD patients. Patients diagnosed with RA and SLE in Fukushima Medical University Hospital were randomly selected. Among 50 patients with RA, 39 (78.0%) were female and their median age was 67 years, [IQR]; 62–74 years. 8 patients (16%) of RA were untreated. Most of the RA patients were taking disease-modifying anti-rheumatic drugs (DMARDs), mostly methotrexate (26/50, 52%), and biologics (17/50, 34%). Median DAS28-ESR was 3.43 (IQR; 2.96–4.03). In addition, 18 (90%) were female Among 20 patients with untreated SLE. Their median age was 44 years, [IQR]; 34.5–52.3 years. The phenotype of SLE included lupus nephritis (8/20, 40%), neuropsychiatric SLE (4/20), and other manifestations.

## Clinical investigation and data collection

In the patient group, clinical, demographic, laboratory features, and medical histories were collected by reviewing electronic medical records. The demographic and clinical characteristics were analyzed as follows; gender, date of birth, age at diagnosis, duration of disease, past or family history of rheumatic diseases, presence of skin rash, arthralgia, arthritis, myalgia, fever characteristics, lymphadenopathy, and visceral involvement (serositis, liver damage). The laboratory data were recorded as follows: leukocyte and thrombocyte counts, hemoglobin, C-reactive protein (CRP), aspartate transaminase, alanine transaminase, lactate dehydrogenase (LDH), ferritin, and markers for hemophagocytosis (hypertriglyceridemia, hypofibrinogenemia, hemophagocytosis in the bone narrow). Disease activity score by Pouchot et al. [15] for AOSD was analyzed in each patient. The maximum score of Pouchot's et al. is 12 points. This score consists of the following 12 manifestations: fever, typical rash, pleuritis, pneumonia, pericarditis, hepatomegaly or abnormal liver function tests, splenomegaly, lymphadenopathy, leukocytosis > 15,000/mm3, sore throat, myalgia, and abdominal pain. Data were collected using a standardized data extraction form. The rheumatologists double-checked the accuracy of these data. Patients were classified as having two disease patterns, the systemic or the articular manifestations as described previously [16].

## ELISA methods

Serum concentrations of CIRP were measured using enzyme-linked immunosorbent assay (ELISA) kits (MBL, Nagoya, Japan; Code No, CY-8103) according to the manufacturer's instruction. Similarly, Serum IL-18 was measured by ELISA Kits (MBL, Nagoya, Japan; Code No. 7620).

## Statistical analysis

Results were non-normally distributed and are presented throughout the manuscript with median and 25–75th centiles [median, IQR] and were compared by Mann-Whitney's U test. Correlations between continuous variables were analyzed by Spearman's rank correlation test. Paired data were analyzed by Wilcoxon signed-rank test. Kruskal–Wallis test was performed for comparisons of continuous variables among the four groups. Post hoc pairwise analyses between two groups were analyzed by Games-Howell test. All data entry and statistical analyses were performed using SPSS Statistics version 22.0 (IBM, Armonk, NY). All statistical analyses were 2-tailed, and statistical significance was defined as $p < 0.05$.

The study was conducted with the approval of the Ethics Committee of Fukushima Medical University (No. 2889). As this was a retrospective study, the need for written informed consent was waived.

# Results

## Clinical characteristics of patients with AOSD

Serum samples were obtained from patients with AOSD. Table 1 summarizes the baseline characteristics and laboratory data from the patients. The principal clinical symptoms of AOSD included a high spiking fever (42/44 95.4%), skin rash (32/44 72.7%), arthralgia (28/44 63.6%), sore throat (14/44 31.8%), and splenomegaly (17/44 38.6%), respectively. After initial investigation, all patients were treated with corticosteroids, and 7 (15.9%) of them also received at least one biologics (Table 1).

## Serum levels of CIRP in patients with AOSD

Serum levels of CIRP were determined by ELISA in patients with AOSD, patients with RA, and HCs. As shown in Fig 1, serum levels of CIRP were significantly higher in patients with AOSD (median: 9.6 ng/mL, IQR [6.1–13.7]) compared to those in patients with RA (3.2 ng/mL, IQR [1.9–3.8]; $p < 0.001$), SLE (2.2 ng/mL, IQR [1.9–2.4]; $p < 0.001$) and in HCs (2.8 ng/mL, [IQR; 1.4–4.9], $p < 0.001$). Serum levels of AOSD patients were significantly higher than that of untreated RA (2.6 ng/mL, [IQR; 1.7–3.8], $p < 0.001$). Three patients with AOSD were complicated with hemophagocytosis syndrome. There was no significant difference in serum levels of CIRP among patients with AOSD with and without hemophagocytosis syndrome (median: 7.3 ng/mL, IQR [6.2–22.0] versus median: 9.6 ng/mL, IQR [5.9–13.9] $p = 0.92$). We also compared serum levels of CIRP according to the disease activity of AOSD. In the sub-group patients with active AOSD, serum levels of CIRP were significantly higher compared to those in inactive AOSD patients (Fig 2).

**Table 1. Characteristics of AOSD patients.**

| Characteristics | Value |
|---|---:|
| Number, n | 44 |
| age(years),median(IQR) | 41 (31.8–5) |
| Age at onset(years), median(IQR) | 39.5 (27.5–56) |
| Male, n(%) | 14 (32) |
| WBC(/μL), median(IQR) | 10600 (8375–16400) |
| Ferritin(ng/mL), median(IQR) | 1015 (322–3878) |
| CRP(mg/dL), median | 5.45 (2.9–10.3) |
| ALT(IU/L), median(IQR) | 33 (20.5–63.5) |
| IL-18 (pg/mL), median(IQR) | 40247 (11620–128209) |
| CIRP (ng/mL), median(IQR) | 9.62 (6.05–13.71). |
| Pouchot's score, median(IQR) | 3 (2–4.3) |
| PSL(mg/day), median(IQR) | 40 (40–60) |
| Corticosteroid pulse therary, n(%) | 22 (50) |
| Immunosuppressant, n(%) | 32 (72.7) |
| Biologics, n(%) | 7 (15.9) |
| Polycyclic systemic type, n(%) | 23 (52.2) |
| Monocyclic systemic type, n(%) | 15 (34.1) |
| Chronic arthritis type, n(%) | 6 (13.6) |

AOSD = adult-onset Still's disease, WBC = white blood cell, CIRP = Cold inducible RNA-binding protein, CRP = C reactive protein, ALT = alanine aminotransferase, PSL = prednisolone, IQR = interquartile range

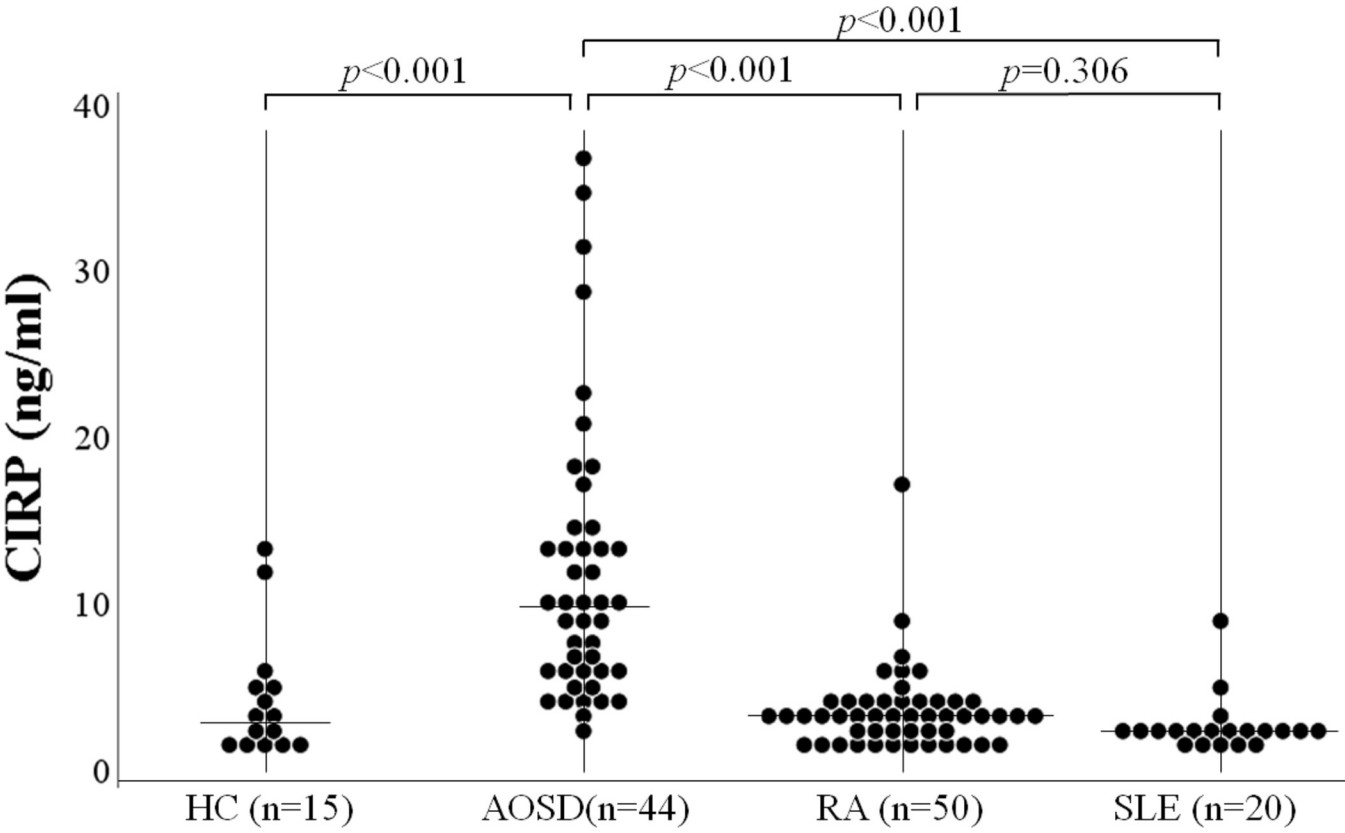

**Fig 1. Serum levels of CIRP in patients with AOSD.** Serum levels of CIRP in AOSD patients (n = 44) were significantly higher compared to those in RA patients (n = 50), SLE patients (n = 20) or healthy controls (n = 15). Comparisons of serum CIRP levels by Kruskal–Wallis test is significantly different among the four groups ($p<0.001$). Post hoc pairwise analyses between two groups were analyzed by Games-Howell test.

### Relationship between serum levels of CIRP and laboratory parameters in patients with AOSD

The correlation between serum levels of CIRP and laboratory parameters were evaluated in patients with AOSD. As illustrated in Fig 3A, serum levels of CIRP showed a significant correlation with serum ferritin levels ($r = 0.47$, $p = 0.002$), but not with CRP levels (Fig 3B). As shown in Fig 3C, serum levels of CIRP were positively correlated with the AOSD disease activity score (Pouchot's score $r = 0.46$, $p = 0.003$). Positive correlation was also demonstrated between serum levels of CIRP and IL-18 ($r = 0.33$, $p = 0.03$) in patients with AOSD (Fig 3D). To determine whether serum CIRP could be used to differentiate AOSD phenotypes, we compared serum levels of CIRP among three phenotypes of AOSD. However, there was no significant difference in serum levels of CIRP among AOSD patients with three different phenotypes (Fig 4).

Kruskal–Wallis test was used for continuous variables for comparisons between three groups.

### Longitudinal observation of serum levels of CIRP

To explore the longitudinal changes of CIRP, we included 8 AOSD patients with two longitudinal samples (at least one month apart). In the longitudinal study, eight patients with active AOSD were followed until they became inactive and then resampled. Serum levels of CIRP

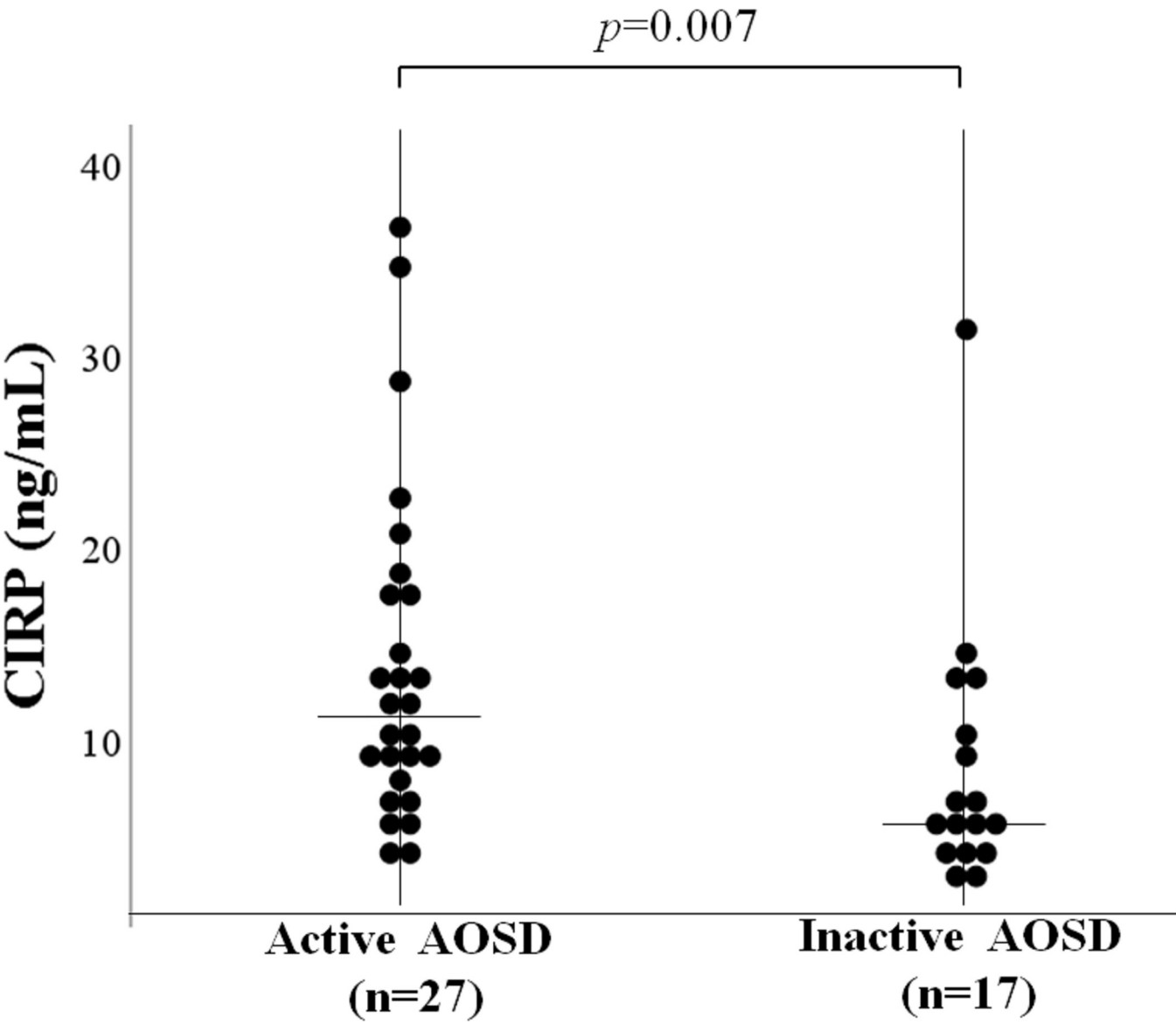

**Fig 2. Serum levels of CIRP levels between active and inactive AOSD patients.** Serum levels of CIRP were significantly higher in active AOSD patients than those in inactive AOSD patients. Inactive AOSD is defined when Pouchot's score is below 3 points. Results were presented with median and were compared by the Mann-Whitney U test.

significantly declined in patients with AOSD after immunosuppressive therapies (Fig 5A), paralleling to the AOSD disease activity score (Fig 5B) as well as to serum levels of ferritin (Fig 5C).

## Discussion

This is the first known study investigating the serum levels of CIRP in autoinflammatory disease, AOSD. Based on the current understanding of the molecular pathogenesis of AOSD, a genetic background would confer the activation of innate immunity in response to environmental factors [12]. DAMPs can activate innate immunity [17], and it was hypothesized that changes in several DAMPs are associated with AOSD and mediate autoinflammatory cascade by activation of inflammasome [13]. CIRP is induced by cellular stresses and functions as a

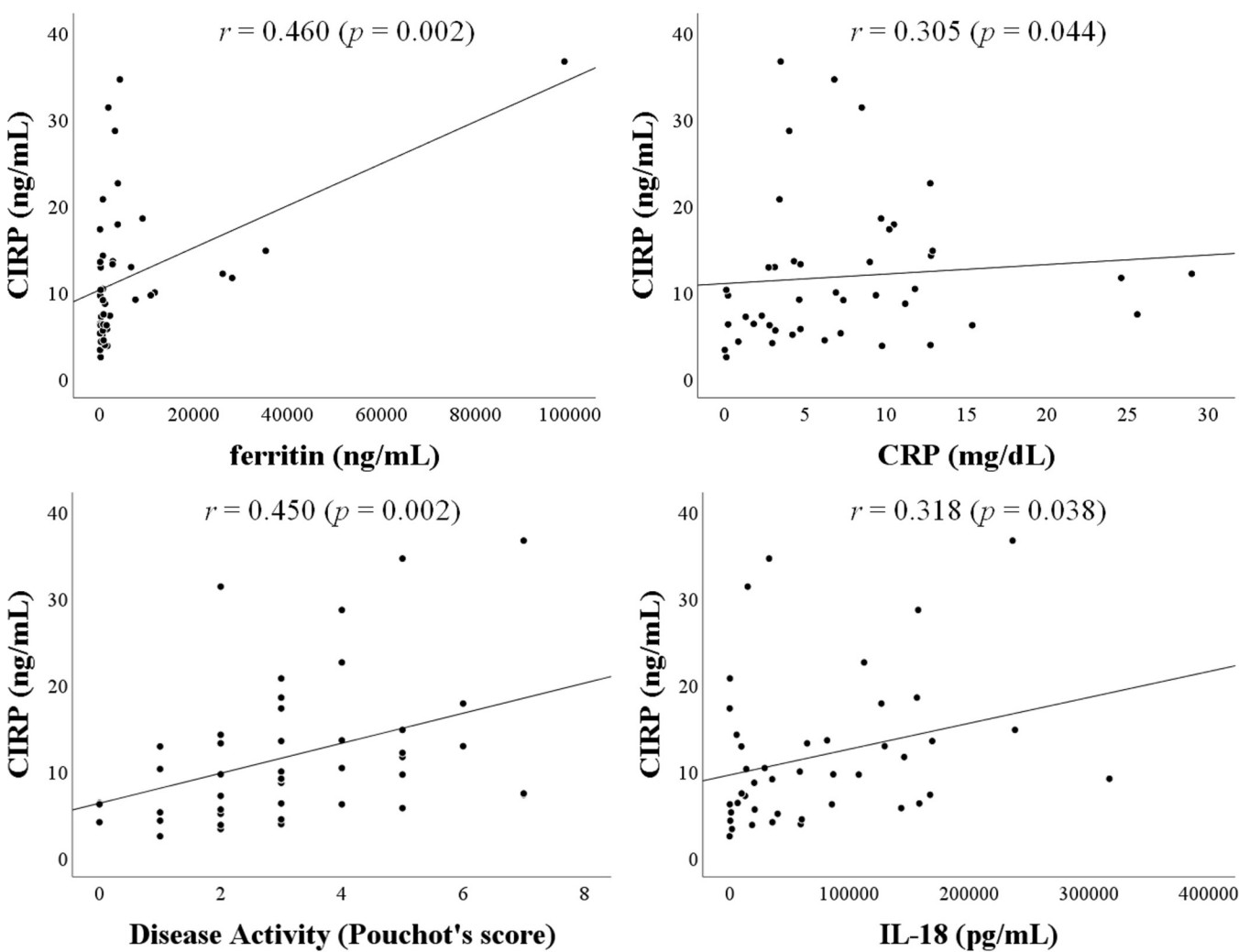

**Fig 3. Relationship between serum levels of CIRP and clinical parameters in patients with AOSD. A.** Correlation analysis of serum levels of CIRP and ferritin showed a significant positive correlation in AOSD patients. **B.** Correlation analysis of serum levels of CIRP and CRP did not show a significant correlation in AOSD patients. **C.** Correlation analysis of serum levels of CIRP and disease activity scores (Pouchot's score) showed a significant positive correlation in AOSD patients. **D.** Correlation analysis of serum levels of CIRP and IL-18 showed a significant positive correlation in AOSD patients.

DAMP molecule that promotes inflammatory responses [4]. We found that extracellular serum levels of CIRP, which is an endogenous DAMP, were elevated in patients with AOSD. Increased serum and synovial fluid levels of CIRP were also reported in patients with RA and osteoarthritis, and the increased synovial levels of CIRP correlated with disease activity of RA [18, 19]. However, there was no significant difference in serum levels of CIRP between patients with RA, SLE, and healthy controls in our study. In contrast, patients with AOSD had significantly higher serum CIRP levels compared with those in patients with RA and SLE. The serum levels of CIRP were positively correlated with Pouchot's score, a disease activity score for AOSD. Furthermore, the elevated levels of CIRP correlated with serum levels of IL-18, a signature cytokine for AOSD [20]. Our data expanded the role of CIRP in an autoinflammatory disorder, AOSD.

Intracellular CIRP (iCIRP) and eCIRP may have different functions. iCIRP has its role in regulating cellular stress responses, such as mRNA stability, cell proliferation cell survival, and tumor formation [21]. In contrast to iCIRP, eCIRP is presumed to function as a DAMP-

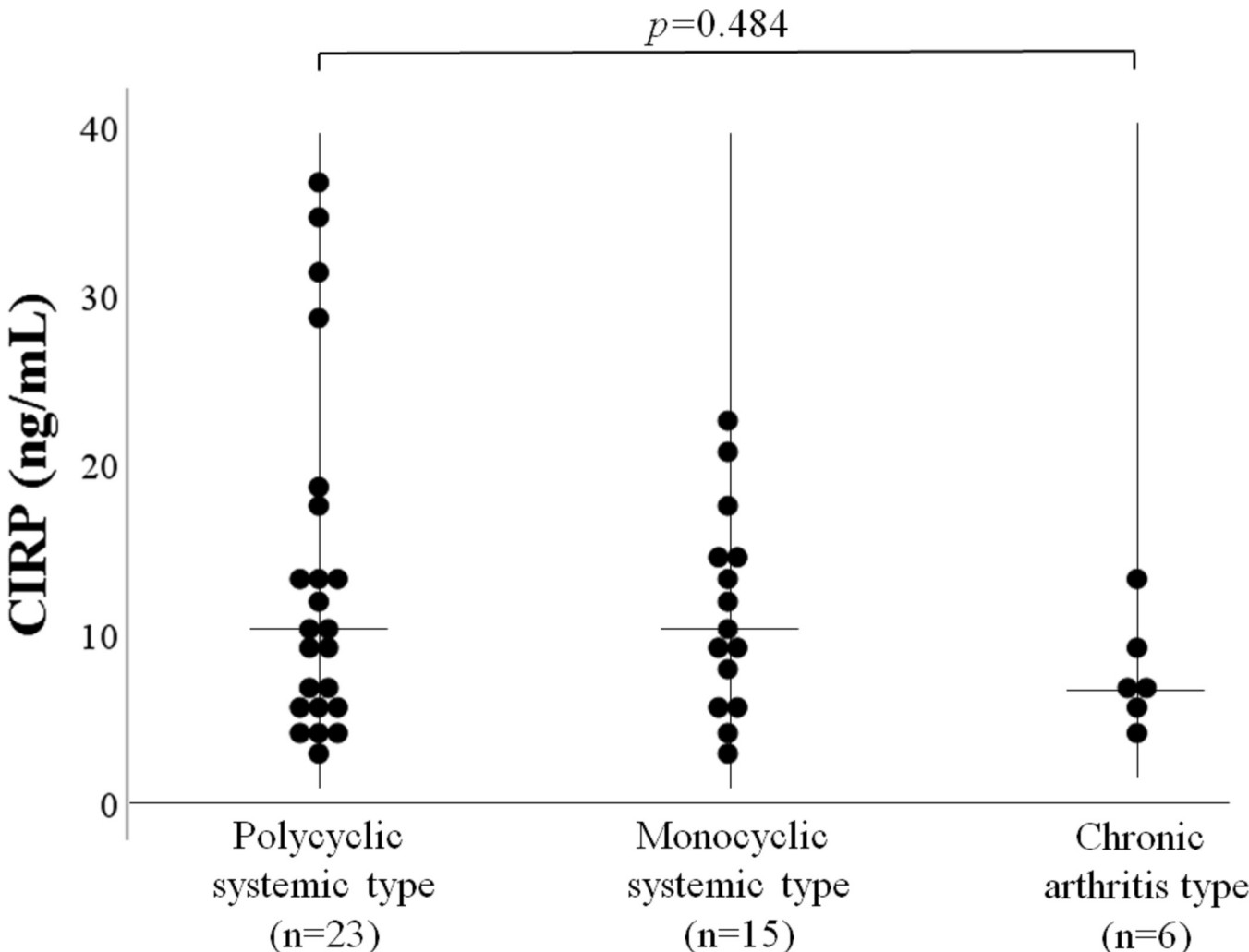

**Fig 4. Serum levels of CIPR in AOSD patients with three different phenotypes.** We compared serum levels of CIRP among AOSD patients with three different disease phenotypes. There was no significant difference in serum levels of CIRP among AOSD patients with three different disease phenotypes.

enhancing inflammation and tissue injury [22]. eCIRP was able to induce proinflammatory cytokines through activating innate immune cells [9]. For example, macrophage treated with recombinant CIRP releases TNF-α and HMGB-1 [23]. Moreover, previous studies demonstrated that CIRP induces the activation of NLRP3 inflammasome [24], resulting in the release of IL-1β, a critical cytokine in AOSD. Dysregulated NLRP3 inflammasome activation is associated with hereditary autoinflammatory diseases as well as with the acquired autoinflammatory diseases, AOSD [25]. It is possible, therefore, that elevated serum CIRP may activate NLRP3 inflammasome and subsequent activated IL-1β induction implicated in the pathogenesis of AOSD.

NLRP3, a pattern recognition receptor, is activated by DAMPs released from cells during cellular stress [26]. In AOSD, PAMPs or DAMPs, in response to infections or environmental factors, transmit to innate immune cells through pattern recognition receptors which activate the NLRP3 inflammasome under the predisposing genetic background [12]. The underlying mechanism by which CIRP-induced inflammation has been proposed, in which eCIRP could be an endogenous DAMP that triggers autoinflammation [27].

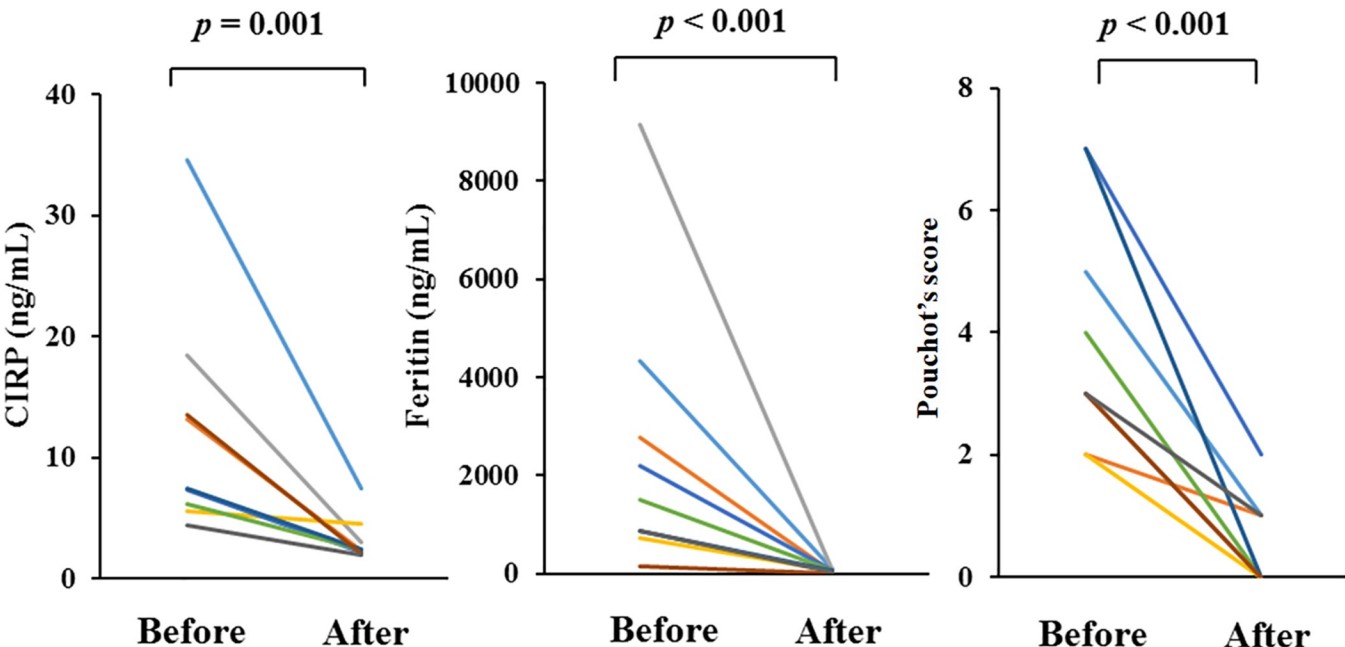

**Fig 5.** Longitudinal changes of serum levels of CIRP (A), ferritin (B) and disease activity score (Pouchot's score) in 8 patients with active AOSD before and after immunosuppressive treatments. Paired samples from the same subjects (n = 8) were compared by Wilcoxon signed-rank test.

More recently, Denning et al. demonstrated that eCIRP is an endogenous triggering receptor expressed on myeloid cells-1 (TREM-1) ligand and their interaction fuels inflammation [28]. Furthermore, serum levels of soluble TREM-1 (sTREM-I) are elevated and correlated with disease activity of patients with AOSD [29]. Although the functions of sTREM-1 were not completely elucidated, sTREM-I is thought to negatively regulate TREM-I signaling. Therefore, it is possible that elevated circulating CIRP may be related to the activation of TREM-I in AOSD patients. Additional studies are warranted to confirm the mechanism through which CIRP could mediate autoinflammation seen in AOSD.

This study has limitations. First, it is difficult to determine the association between extracellular (serum) and intracellular CIRP expressions because the functional analysis in vitro cannot be performed. Second, we conducted a single center retrospective cohort study with a small sample size of patients; more studies based on larger cohort in additional sites are necessary to verify our findings.

## Conclusions

Serum concentrations of CIRP were elevated in patients with AOSD. Furthermore, serum concentrations of CIRP correlated with serum levels of ferritin and disease activity score in patients with AOSD. Our results suggest that CIRP may be implicated in the pathophysiology of AOSD and be a potential marker for autoinflammation seen in AOSD. These data provided us new insights into the role of CIRP in AOSD and highlighted the therapeutic potential targeting of CIRP in the regulation of autoinflammation.

## Supporting information

**S1 Dataset.**
(DOCX)

## Acknowledgments

We are grateful to Ms. Kanno Sachiyo for her technical assistance in this study.

## Author Contributions

**Conceptualization:** Yuya Fujita, Toru Yago, Atsushi Kawakami, Kiyoshi Migita.

**Data curation:** Yuya Fujita, Tomoyuki Asano, Haruki Matsumoto, Naoki Matsuoka, Jumpei Temmoku, Makiko Yashiro-Furuya, Eiji Suzuki, Hiroshi Watanabe.

**Formal analysis:** Yuya Fujita.

**Investigation:** Yuya Fujita, Kiyoshi Migita.

**Methodology:** Yuya Fujita, Toru Yago, Kiyoshi Migita.

**Supervision:** Atsushi Kawakami, Kiyoshi Migita.

**Validation:** Haruki Matsumoto, Shuzo Sato, Kiyoshi Migita.

**Writing – original draft:** Yuya Fujita, Kiyoshi Migita.

**Writing – review & editing:** Yuya Fujita, Toru Yago, Kiyoshi Migita.

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
