## [Decision Letter · Decision Letter 0]

20 Apr 2021

PONE-D-21-02804

Clinical relevance for circulating cold inducible RNA-binding protein (CIRP) in patients with adult-onset Still’s disease

PLOS ONE

Dear Dr. Fujita,

Thank you for submitting your manuscript to PLOS ONE. After careful consideration, we feel that it has merit but does not fully meet PLOS ONE’s publication criteria as it currently stands. Therefore, we invite you to submit a revised version of the manuscript that addresses the points raised during the review process.

We look forward to receiving your revised manuscript.

Kind regards,

Manal S. Fawzy, Ph.D., M.D.

Academic Editor

PLOS ONE

2. Please provide additional details regarding participant consent. In the ethics statement in the Methods and online submission information, please ensure that you have specified:

 - whether consent was obtained

 - whether consent was informed

 - what type of consent you obtained (for instance, written or verbal, and if verbal, how it was documented and witnessed).

 - if your study included minors, state whether you obtained consent from parents or guardians.

 - if the need for consent was waived by the ethics committee, please include this information.

3. In your Methods section, please provide additional information about the participant recruitment method and the demographic details of your participants. Please ensure you have provided sufficient details to replicate the analyses such as:

a) a statement as to whether your sample can be considered representative of a larger population, and

b) a description of how participants from each group were recruited.

4. Thank you for providing the date(s) when patient medical information was initially recorded. Please also include the date(s) on which your research team accessed the databases/records to obtain the retrospective data used in your study.

5. Please provide the catalog numbers of the ELISA kits used in the study.

6. Thank you for stating the following in the Competing Interests section:

"Kiyoshi Migita has received research grants from Chugai, Pfizer, and AbbVie. Rest of the authors declares that they have no competing interests."

7. Thank you for submitting the above manuscript to PLOS ONE. During our internal evaluation of the manuscript, we found significant text overlap between your submission and the following previously published works.

- https://doi.org/10.1186/s13075-020-02263-3

- https://doi.org/10.1002/JLB.3MIR1118-443R

We would like to make you aware that copying extracts from previous publications, especially outside the methods section, word-for-word is unacceptable, even for works which you authored. In addition, the reproduction of text from published reports has implications for the copyright that may apply to the publications.

Please revise the manuscript to rephrase the duplicated text, cite your sources, and provide details as to how the current manuscript advances on previous work. Please note that further consideration is dependent on the submission of a manuscript that addresses these concerns about the overlap in text with published work.

Reviewers' comments:

Reviewer's Responses to Questions

**Comments to the Author**

1. Is the manuscript technically sound, and do the data support the conclusions?

Reviewer #1: Yes

Reviewer #2: Partly

2. Has the statistical analysis been performed appropriately and rigorously? 

Reviewer #1: Yes

Reviewer #2: No

3. Have the authors made all data underlying the findings in their manuscript fully available?

Reviewer #1: Yes

Reviewer #2: Yes

4. Is the manuscript presented in an intelligible fashion and written in standard English?

Reviewer #1: Yes

Reviewer #2: No

5. Review Comments to the Author

Reviewer #1: 1. The age of the control, AOSD and RA groups were significantly different, which may affect immune response.

2. In the method section, please explain the drawing time of the laboratory data as well as the drawing time for CIRP before and after treatment.

3. Please add IL-18 in table 1

4. Fig 1 please confirm the number of patients in each group

5. Some writing error: such as: figures, ASD should be AOSD

Reviewer #2: This article explored the clinical significance of serum CIRP levels in AOSD for the first time. According to the article, the serum CIRP levels seem to significantly be elevated in AOSD patients and be related to the disease activity of AOSD.

Strengths:

CIRP was a ligand to TREM-1 and the interaction of them could aggravate inflammatory responses, thus, the elevated serum CIRP may related to the pathogenesis of AOSD.

Weaknesses:

1. The sample size of this study is relatively small.

2. They only included RA as disease control, but the differential diagnosis of AOSD should be made among infectious diseases, hematological malignancy etc.

6. PLOS authors have the option to publish the peer review history of their article (what does this mean?). If published, this will include your full peer review and any attached files.

Reviewer #1: No

Reviewer #2: No

---

## [Author Response · Author response to Decision Letter 0]

9 Jun 2021

Response to Reviewer #1 

We wish to express our appreciation to your insightful comments on our paper. The comments have helped us significantly improve the paper.

Comment

1. The age of the control, AOSD and RA groups were significantly different, which may affect immune response.

Response

I appreciate for your critical comment. The association between CIRP and age has not been reported. I added systemic lupus erythematosus (SLE) patients as control. The age between AOSD and SLE were equal and serum levels of CIRP in AOSD patients were significantly higher than that in SLE patients. Therefore, I consider the age did not affect immune response.

Comment

2. In the method section, please explain the drawing time of the laboratory data as well as the drawing time for CIRP before and after treatment.

Response

I appreciate for important your comment. In accordance with your comment, I added as follows; “A total of 44 untreated AOSD patients and 15 healthy controls (HCs) were included in the training and validation sets.” “We additionally collected serum samples from 8 treated AOSD patients at the time of remission in order to explore the longitudinal changes.”

Comment

3. Please add IL-18 in table 1

Response

I appreciate for important your comment. In accordance with your comment, I added IL-18 in table 1.

Comment

4. Fig 1 please confirm the number of patients in each group

Response

I appreciate for your comment. I remake correctly figure 1 because of the number of patients was wrong.

Comment

5. Some writing error: such as: figures, ASD should be AOSD

Response

I appreciate for your important comment. I unified the abbreviation of adult onset Still’s disease to AOSD in this manuscript.

Response to Reviewer 2

We wish to express our appreciation to your insightful comments on our paper. The comments have helped us significantly improve the paper.

Comment

This article explored the clinical significance of serum CIRP levels in AOSD for the first time. According to the article, the serum CIRP levels seem to significantly be elevated in AOSD patients and be related to the disease activity of AOSD.

Strengths:

CIRP was a ligand to TREM-1 and the interaction of them could aggravate inflammatory responses, thus, the elevated serum CIRP may related to the pathogenesis of AOSD.

Weaknesses:

1. The sample size of this study is relatively small.

Response

I appreciate for your critical comment. Since AOSD was rare disease, the sample size of this study was relatively small. I added two newly diagnosed patients. Therefore, the number of patients was increased from 42 to 44. 

Comment

2. They only included RA as disease control, but the differential diagnosis of AOSD should be made among infectious diseases, hematological malignancy etc.

Response

I appreciate for your critical comment. As you pointed out, it is important to compare the multiple diseases since the differential diagnosis of AOSD is diverse. I could not collect enough number of infectious diseases and hematological malignancy patients. However, I added systemic lupus erythematosus patients as control. Since AOSD and SLE cause hemophagocytic syndrome, the different diagnosis between AOSD and SLE is especially important. This additional study revealed that serum levels of CIRP in AOSD patients were significantly higher than that in SLE patients. Therefore, I added as follows; “As shown in Fig 1, serum levels of CIRP were significantly higher in patients with AOSD (median: 9.6 ng/mL, IQR [6.1–13.7]) compared to those in patients with RA (3.2 ng/mL, IQR [1.9–3.8]; p < 0.001), SLE (2.2 ng/mL, IQR [1.9–2.4]; p < 0.001) and in HCs (2.8 ng/mL, [IQR; 1.4–4.9], p < 0.001).”

Editor Comments

Comment

Response

I appreciate for your important comment. I ensured our manuscript meets PLOS ONE's style requirements, and revised manuscript exactly.

Comment

2. Please provide additional details regarding participant consent. In the ethics statement in the Methods and online submission information, please ensure that you have specified:

 - whether consent was obtained

 - whether consent was informed

 - what type of consent you obtained (for instance, written or verbal, and if verbal, how it was documented and witnessed).

 - if your study included minors, state whether you obtained consent from parents or guardians.

 - if the need for consent was waived by the ethics committee, please include this information.

Response

I appreciate for your critical comment. In accordance with your comment, I added details regarding participant consent as follows in methods section; “The study was conducted with the approval of the Ethics Committee of Fukushima Medical University (No. 2889). As this was a retrospective study, the need for written informed consent was waived.”

Comment 

3. In your Methods section, please provide additional information about the participant recruitment method and the demographic details of your participants. Please ensure you have provided sufficient details to replicate the analyses such as:

a) a statement as to whether your sample can be considered representative of a larger population, and

b) a description of how participants from each group were recruited.

Response

I appreciate for you essential comment. Patients from wide range of Fukushima prefecture were recruited. Therefore, I think our sample can be considered representative of larger population.

I added the description how to recruit the patients in methods section as follows; “All patients diagnosed with AOSD at the Department of Rheumatology, Fukushima Medical University Hospital from January 2005 to April 2020 were enrolled. All records were accessed from January 2020 to October 2020.” 

“HCs lacked chronic medical diseases or conditions and did not take prescription medications or over-the-counter medications within seven days.” 

“Patients diagnosed with RA and SLE in Fukushima Medical University Hospital were randomly selected.”

Comment

4. Thank you for providing the date(s) when patient medical information was initially recorded. Please also include the date(s) on which your research team accessed the databases/records to obtain the retrospective data used in your study.

Response

I appreciate for your important comment. In accordance with your comment, we added as follows in methods section” All patients diagnosed with AOSD at Department of Rheumatology, Fukushima Medical University Hospital from January 2005 to April 2020 were enrolled. All records were accessed from January 2020 to October 2020.”

Comment

5. Please provide the catalog numbers of the ELISA kits used in the study.

Response

I appreciate for your comment. I added the catalog numbers of the ELISA kits as followed in method section; “Serum concentrations of CIRP� were measured using enzyme-linked immunosorbent assay (ELISA) kits (MBL, Nagoya, Japan; Code No, CY-8103) according to the manufacturer's instruction. Similarly, Serum IL-18 was measured by ELISA Kits (MBL, Nagoya, Japan; Code No. 7620).”

Comment

6. Thank you for stating the following in the Competing Interests section:

"Kiyoshi Migita has received research grants from Chugai, Pfizer, and AbbVie. Rest of the authors declares that they have no competing interests."

Response

I appreciate for your important comment. I confirmed this does not alter our adherence to PLOS ONE policies on sharing data and materials. Therefore I included our updated Competing Interests statement in our cover letter as follews;” Kiyoshi Migita has received research grants from Chugai, Pfizer, and AbbVie. Rest of the authors declares that they have no competing interests. This does not alter our adherence to PLOS ONE policies on sharing data and materials.”

Comment

7. Thank you for submitting the above manuscript to PLOS ONE. During our internal evaluation of the manuscript, we found significant text overlap between your submission and the following previously published works.

- https://doi.org/10.1186/s13075-020-02263-3

- https://doi.org/10.1002/JLB.3MIR1118-443R

We would like to make you aware that copying extracts from previous publications, especially outside the methods section, word-for-word is unacceptable, even for works which you authored. In addition, the reproduction of text from published reports has implications for the copyright that may apply to the publications.

Please revise the manuscript to rephrase the duplicated text, cite your sources, and provide details as to how the current manuscript advances on previous work. Please note that further consideration is dependent on the submission of a manuscript that addresses these concerns about the overlap in text with published work.

Response

I appreciate for your critical comment. I confirmed previous publish works pointed out by editors, and I rephrase the duplicated text.

---

## [Decision Letter · Decision Letter 1]

7 Jul 2021

PONE-D-21-02804R1

Clinical relevance for circulating cold-inducible RNA-binding protein (CIRP) in patients with adult-onset Still’s disease

PLOS ONE

Dear Dr. Fujita,

Thank you for submitting your manuscript to PLOS ONE. After careful consideration, we feel that it has merit but does not fully meet PLOS ONE’s publication criteria as it currently stands. Therefore, we invite you to submit a revised version of the manuscript that addresses the points raised during the review process.

We look forward to receiving your revised manuscript.

Kind regards,

Manal S. Fawzy, Ph.D., M.D.

Academic Editor

PLOS ONE

Journal Requirements:

Additional Editor Comments (if provided):

Please address the raised concern by the reviewer and highlight in the revised version of the manuscript.

Reviewers' comments:

Reviewer's Responses to Questions

**Comments to the Author**

1. If the authors have adequately addressed your comments raised in a previous round of review and you feel that this manuscript is now acceptable for publication, you may indicate that here to bypass the “Comments to the Author” section, enter your conflict of interest statement in the “Confidential to Editor” section, and submit your "Accept" recommendation.

Reviewer #1: All comments have been addressed

Reviewer #2: All comments have been addressed

2. Is the manuscript technically sound, and do the data support the conclusions?

Reviewer #1: Yes

Reviewer #2: Partly

3. Has the statistical analysis been performed appropriately and rigorously? 

Reviewer #1: Yes

Reviewer #2: No

4. Have the authors made all data underlying the findings in their manuscript fully available?

Reviewer #1: Yes

Reviewer #2: Yes

5. Is the manuscript presented in an intelligible fashion and written in standard English?

Reviewer #1: Yes

Reviewer #2: Yes

6. Review Comments to the Author

Reviewer #1: The authors answered all questions well. In my opinion, I do not think I can give some suggestions for current version.

Reviewer #2: The authors added SLE and RA as disease control, but the medication seems diverse in RA patients (the medication of SLE patients not described). The influence of medication such as glucocorticoids may also affect the level of CIRP.

7. PLOS authors have the option to publish the peer review history of their article (what does this mean?). If published, this will include your full peer review and any attached files.

Reviewer #1: No

Reviewer #2: No

---

## [Author Response · Author response to Decision Letter 1]

10 Jul 2021

Response to Reviewer #1 

Comment

The authors answered all questions well. In my opinion, I do not think I can give some suggestions for current version.

Response

I am grateful for your comment

Response to Reviewer 2

We wish to express our appreciation to your insightful comments on our paper. The comments have helped us significantly improve the paper.

Comment

The authors added SLE and RA as disease control, but the medication seems diverse in RA patients (the medication ofSLE patients not described). The influence of medication such as glucocorticoids may also affect the level of CIRP.

Response

I appreciate for your critical comment. All SLE patients was untreated. Among 50 RA patients, 8 RA patients was untreated. We added the comparison between untreated AOSD and untreated RA as follows, “Serum levels of AOSD patients were significantly higher than that of untreated RA (2.6 ng/mL, [IQR; 1.7-3.8], p<0.001).”

---

## [Decision Letter · Decision Letter 2]

19 Jul 2021

Clinical relevance for circulating cold-inducible RNA-binding protein (CIRP) in patients with adult-onset Still’s disease

PONE-D-21-02804R2

Dear Dr. Fujita,

We’re pleased to inform you that your manuscript has been judged scientifically suitable for publication and will be formally accepted for publication once it meets all outstanding technical requirements.

Kind regards,

Manal S. Fawzy, Ph.D., M.D.

Academic Editor

PLOS ONE

Additional Editor Comments (optional):

The authors have adequately addressed the concerns raised by the reviewers.

Reviewers' comments:

Reviewer's Responses to Questions

**Comments to the Author**

1. If the authors have adequately addressed your comments raised in a previous round of review and you feel that this manuscript is now acceptable for publication, you may indicate that here to bypass the “Comments to the Author” section, enter your conflict of interest statement in the “Confidential to Editor” section, and submit your "Accept" recommendation.

Reviewer #2: All comments have been addressed

2. Is the manuscript technically sound, and do the data support the conclusions?

Reviewer #2: Yes

3. Has the statistical analysis been performed appropriately and rigorously? 

Reviewer #2: Yes

4. Have the authors made all data underlying the findings in their manuscript fully available?

Reviewer #2: Yes

5. Is the manuscript presented in an intelligible fashion and written in standard English?

Reviewer #2: Yes

6. Review Comments to the Author

Reviewer #2: The authors clarified my concerns well. From my point of view, I have no more questions or suggestions.

7. PLOS authors have the option to publish the peer review history of their article (what does this mean?). If published, this will include your full peer review and any attached files.

Reviewer #2: No

---

## [Editor Report · Acceptance letter]

26 Jul 2021

PONE-D-21-02804R2 

Clinical relevance for circulating cold-inducible RNA-binding protein (CIRP) in patients with adult-onset Still’s disease 

Dear Dr. Fujita:

I'm pleased to inform you that your manuscript has been deemed suitable for publication in PLOS ONE. Congratulations! Your manuscript is now with our production department. 

Kind regards, 

on behalf of

Professor Manal S. Fawzy 

Academic Editor

PLOS ONE